# Correspondence-Oriented Imitation Learning: Flexible Visuomotor Control with 3D Conditioning

## Abstract

We introduce Correspondence-Oriented Imitation Learning (COIL), a conditional policy learning framework for visuomotor control with a flexible task representation in 3D. At the core of our approach, each task is defined by the intended motion of keypoints selected on objects in the scene. Instead of assuming a fixed number of keypoints or uniformly spaced time intervals, COIL supports task specifications with variable spatial and temporal granularity, adapting to different user intents and task requirements. To robustly ground this correspondence-oriented task representation into actions, we design a conditional policy with a spatio-temporal attention mechanism that effectively fuses information across multiple input modalities. The policy is trained via a scalable self-supervised pipeline using demonstrations collected in simulation, with correspondence labels automatically generated in hindsight. COIL generalizes across tasks, objects, and motion patterns, achieving superior performance compared to prior methods on real-world manipulation tasks under both sparse and dense specifications. For additional results and videos, please visit: `https://coil-manipulation.github.io/`

## 1 Introduction

Learning general-purpose policies capable of performing diverse tasks in the physical world requires an interface that can flexibly specify task intent and constraints. In the context of visuomotor control, policies conditioned on high-level task representations, such as goal images (Sundaresan et al., 2024; Nair et al., 2018) and language instructions (Brohan et al., 2022; Jiang et al., 2022; Collaboration, 2023), have been widely studied for a variety of tasks. However, generating precise, executable actions from these representations typically involves grounding them in high-dimensional visual observations, which poses a significant challenge for policy learning. This highlights the need for task representations that can bridge high-level task semantics and low-level physical interactions in a seamless manner.

Recent work has explored the use of motion trajectories of objects and robots, commonly referred to as flows, as an alternative representation of tasks (Gu et al., 2023; Sundaresan et al., 2024; Xu et al., 2024; Wen et al., 2023; Vecerik et al., 2023). Defined directly in the environment, flows offer interpretable and spatially grounded task specifications that facilitate policy conditioning and generalization. Despite these advantages, existing flow-conditioned policies face several critical limitations. Most approaches operate on 2D flows extracted from monocular video, which are inherently ambiguous in depth and sensitive to occlusions. While recent efforts have extended flows into 3D (Gao et al., 2024; Yuan et al., 2024a;b; Zhi et al., 2025), these methods typically rely on handcrafted motion primitives or controllers, limiting their adaptability to new tasks and environments. Moreover, they often require densely annotated flows to provide detailed motion guidance, making them costly and rigid to specify.

To this end, we introduce Correspondence-Oriented Imitation Learning (COIL), a conditional imitation learning framework for general-purpose visuomotor control. At the core of our approach is a novel task representation based on spatio-temporal correspondences of 3D keypoints, which extends flow representations to support flexible specification of manipulation tasks. As shown in Figure 1, COIL defines each task with the intended trajectories of a set of keypoints selected on the observed

Figure 1: We introduce **COIL**, an approach for versatile manipulation conditioned on a correspondence-oriented task representation in 3D. Each task is defined by a set of keypoints annotated on the observed point cloud of scene objects, with task goals and constraints expressed as their intended 3D trajectories. Unlike prior work that assumes a fixed number of keypoints or densely sampled time steps, COIL supports task specifications with variable spatial and temporal granularity, allowing users or planners to adapt the level of detail based on the task's complexity or intent.

point cloud in the scene. Unlike prior work (Xu et al., 2024; Yuan et al., 2024a; Gao et al., 2024), our formulation imposes no constraints on the number of keypoints or the temporal resolution, allowing task specifications to vary in spatial and temporal granularity based on task need or user intent. To robustly execute tasks from such flexible representations, we design a conditional policy that fuses input observations and task representations across space and time using a spatio-temporal attention mechanism. We train this policy via a scalable imitation learning pipeline that generates diverse demonstrations in simulation and labels the corresponding task specifications automatically through hindsight relabeling, enabling fully self-supervised training.

We validate the effectiveness of COIL on a diverse set of robotic manipulation tasks involving both rigid and deformable objects, tool use, and spatial constraints in 3D environments. The learned policy demonstrates strong zero-shot performance across tasks and generalizes robustly under both sparse and dense task specifications, outperforming baseline methods by a significant margin. An in-depth ablation study further highlights the contribution of each component in our framework toward achieving robust and generalizable visuomotor control.

## 2 RELATED WORK

Designing effective task representations is a central challenge in learning generalizable visuomotor policies for robotic control. Goal-conditioned policies have long provided a practical interface by specifying tasks through target states (Kaelbling, 1993; Schaul et al., 2015). These approaches enable scalable self-supervised training via techniques such as hindsight relabeling (Andrychowicz et al., 2017; Fang et al., 2019b; Pong et al., 2020; Ghosh et al., 2019) and curriculum learning (Florensa et al., 2018; Zhang et al., 2020). While many robotic problems can be framed as goal-reaching tasks, this formulation struggles to capture more complex task constraints, and structured goal states are often difficult to define or acquire. Extensions to partially observable environments incorporate visual goal observations (Ding et al., 2019; Eysenbach et al., 2021; Gupta et al., 2019; Ghosh et al., 2019; Nair et al., 2018; Fang et al., 2019a; Nair et al., 2020; Chane-Sane et al., 2021; Nasiriany et al., 2019; Eysenbach et al., 2019; Fang et al., 2023; 2022; Walke et al., 2023; Myers et al., 2023), but these often over-specify scenes in pixel space, entangling goal-relevant features with distractions like background clutter, lighting, or viewpoint variation. Language-conditioned policies, on the other hand, provide an abstract, high-level task interface (Stepputtis et al., 2020; Jang et al., 2022; Brohan et al., 2022; Jiang et al., 2022; Driess et al., 2023; Lin et al., 2023; Black et al., 2024; Lynch & Sermanet, 2020; Lynch et al., 2023; Brohan et al., 2023) by leveraging advances in large language and vision-language models (Radford et al., 2021; OpenAI, 2024). While free-form language instructions can be used to specify a broad range of task semantics, grounding instructions into precise motions in the physical world requires large-scale robot data and often lacks interpretability. In this work, we propose a mid-level task representation that inherits the merits of goal-conditioned learning, while offering higher expressiveness for more diverse and complex object interactions. Unlike image goals or natural language, our representation can be defined directly on 3D visual observations, avoiding the need for learning nuanced spatial grounding from high-level modalities, yet remaining compact enough for humans or high-level planners to specify.

An increasing number of recent works represent task specifications as the motion of keypoints sampled from objects or the robot, commonly referred to as *flows* or *trajectories* (Gu et al., 2023; Sundaresan et al., 2024; Xu et al., 2024; Wen et al., 2023; Vecerik et al., 2023; Haldar & Pinto, 2025). These flow-conditioned policies offer interpretable and spatially grounded representations and support cross-embodiment generalization. However, most existing approaches are limited to 2D flows derived from monocular video, making them sensitive to occlusions and inherently ambiguous in depth. While a few recent methods extend flow representations into 3D (Gao et al., 2024; Yuan et al., 2024a; Zhi et al., 2025), they rely on trajectory optimization or handcrafted primitives for control. Several approaches also assume flows defined on the end-effector or require end-effector-centric trajectories (Vecerik et al., 2023; Haldar & Pinto, 2025), limiting the representation's generality across embodiments. Moreover, existing systems typically require densely sampled flows over fixed intervals or flows generated through human demonstrations, making them costly to annotate and inflexible for task specification. We extend this idea into a more flexible formulation based on spatio-temporal correspondences of 3D keypoints, and design a flow-matching policy Lipman et al. (2023); Liu (2022) to predict actions conditioned on such a representation. Our correspondence-oriented representation supports specification of arbitrary spatial and temporal granularity, and enables robust control for versatile robotic manipulation.

## 3 METHOD

We present Correspondence-Oriented Imitation Learning (COIL), a framework for learning generalist visuomotor skills through flexible specifications based on spatio-temporal correspondence in 3D space. As shown in Figure 2, the robot receives point clouds and proprioception information at each timestep $t$, denoted as $o_t \in \mathcal{O}$, and executes actions $a_t \in \mathcal{A}$. Our goal is to learn a generalist conditional policy that robustly interprets diverse manipulation specifications and adapts to varying spatial and temporal constraints.

In this section, we will first introduce a novel correspondence-oriented task representation (Section 3.1). Then, we will describe the key design options in our framework for robustly predicting actions by combining information across space and time (Section 3.2). Lastly, we will explain an imitation learning algorithm to efficiently learn the policy through self-supervision (Section 3.3).

### 3.1 CORRESPONDENCE-ORIENTED TASK REPRESENTATION

We introduce a correspondence-oriented task representation to flexibly specify manipulation goals and constraints in 3D space. Extending object-centric flow formulations (Yuan et al., 2024a; Xu et al., 2024), our task representation describes the desired changes of the environment state using a collection of $K$ keypoints selected on the scene objects. As object poses and states change in the environment, the 3D coordinates of these keypoints move accordingly. For each keypoint $k$ ($0 \leq k \leq K-1$), we specify its target 3D position at $H$ discrete steps, yielding the task representation as a tensor $c \in \mathbb{R}^{H \times K \times 3}$, with the slice $c_0$ denoting the initial keypoint positions. This tensor captures the spatio-temporal correspondences of the keypoints, constraining up to $3K$ degrees of freedom of the environment state at each step. Solving the correspondence-oriented tasks requires the robot to move the keypoints to reach the sequence of target coordinates through interactions with the environment.

To enable flexible task specifications, we introduce key extensions that improve the expressiveness and adaptability of the correspondence-oriented specification. In contrast to prior work (Yuan et al., 2024a; Xu et al., 2024), our formulation removes the rigid assumptions of fixed keypoint counts and dense time steps and enables:

- **Spatial flexibility:** Any $K \geq 1$ keypoints can be selected for the task representation $c$ to exert constraints of varying DoFs based on the task requirement. Moreover, the keypoints can be selected on multiple objects, either static or dynamic, allowing specifications of multistage tasks as well as constraints for objects that should stay still.
- **Temporal flexibility:** Similarly, the target time steps are not restricted to fixed-length or evenly spaced intervals. We have $H \geq 2$ target coordinates for each of the $K$ keypoints, allowing users to provide any specifications from sparse start–goal pairs to dense flows. Instead of rigidly reaching

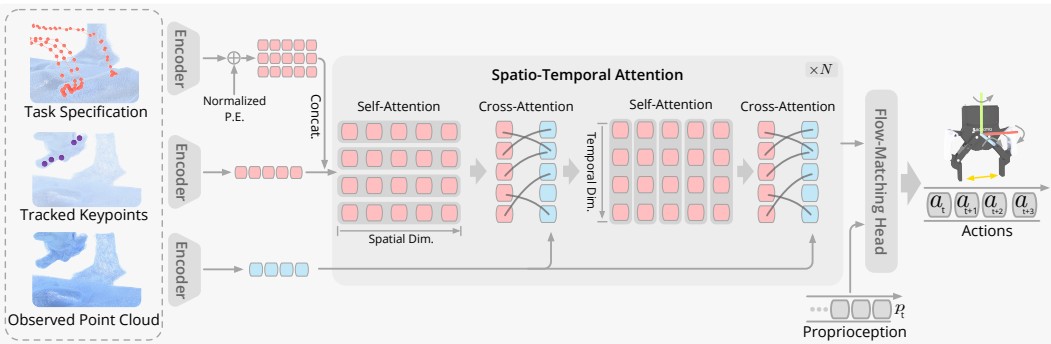

Figure 2: **Overview of COIL Policy.** Our policy encodes the task representation, tracked keypoints, and observed point cloud using shared 3D coordinate encoders. Temporal information is injected via normalized positional encodings. A Spatio-Temporal Transformer efficiently fuses these inputs by interleaving spatial and temporal self-attention and applying cross-attention with the visual observations. The resulting representation is combined with proprioception and passed to a flow-matching head to generate multi-step actions. This design enables effective grounding of task specifications of varying spatial and temporal granularities into precise, executable actions.

the $H$ target coordinates at $H$ predetermined timesteps, we require only that the targets be reached in order, leaving execution speed and recovery behavior free to adapt online.

These extensions enable the correspondence-oriented task representation to specify diverse and complex tasks with constraints at varying spatial and temporal granularities through a unified interface.

## 3.2 ACTION PREDICTION WITH SPATIO-TEMPORAL ATTENTION

We learn a conditional policy $\pi$ to perform the tasks specified through the correspondence-oriented interface. At each time step $t$, $\pi(a_{t:t+T_a-1}|o_{0:t}, c)$ maps the observation history $o_{0:t}$ and the task representation $c$ to the distribution of an action sequence $a_{t:t+T_a-1}$ of a prediction horizon $T_a$, similar to Chi et al. (2024) but with the flow-matching objective described in Section A.2.

The correspondence-oriented action prediction problem can be solved by adapting a heuristic algorithm similar to path following Doersch et al. (2024). But in contrast to classical path following, which usually assumes known states and dynamics, we need to handle uncertainty in observation and dynamics. To robustly interpret and execute the tasks specified by $c$, we decompose action prediction into three stages: (1) tracking the current 3D coordinates of task keypoints $u_t \in \mathbb{R}^{K \times 3}$, (2) grounding the task representation $c$ in the observed environment, and (3) generating actions based on the grounded task understanding. The first problem can be solved explicitly using standard point tracking (Doersch et al., 2024), given the initial keypoint positions $c_0$ and observation history $o_{0:t}$. Once the tracked coordinates $u_t$ are obtained, we aim to locate the task progress by computing step index $j(t)$ for the next target. However, due to noise introduced by point-tracking models and sparse information contained in the task specification, naively grounding the step index $j(t)$ using closest distance projection can be undesirable. Instead, we ask the policy to ground the task progress and predict actions that advance the keypoints toward the remaining targets with priors learned from the training data without explicitly computing $j(t)$.

To effectively overcome the challenge of sparse task specifications and noisy tracking in the physical world, we introduce a spatio-temporal encoder $f$ outlined in Figure 2, that fuses information from observed point cloud $x_t$, tracked keypoints $u_t$, and task representation $c$ across space and time. The resulting embedding is passed to a flow-matching prediction head (Lipman et al., 2023) with a UNet architecture similar to Chi et al. (2024) to model multimodal action distributions, which is particularly important when coarsely-grained correspondences are used to specify the task.

We denote the policy with the encoder $f$ as $\pi(a_{t:t+T_a-1} \mid f(x_t, \rho_t, c_{t:H}))$, and further simplify the notation to $\pi(a \mid f(x, \rho, c))$ for clarity in the remainder of this section. Here we describe the key components of the encoder $f$ that enable effective information fusion for action prediction from flexible correspondence specifications.

**3D encoding.** We first separately process the spatial information of the point cloud $x_t$, the correspondence representations $c$, and the tracked points $u_t$. These three input modalities are all 3D coordinates defined in the same coordinate system, with the difference being that whether they are directly observed from the environment, specified in the task representation, or estimated through point tracking. Therefore, we use different encoders for each of these three input modalities, but share network weights within them to reduce computational complexity. For $u_t \in \mathbb{R}^{K \times 3}$, we apply the same MLP independently to each of the $K$ keypoint positions. While for $c \in \mathbb{R}^{H \times K \times 3}$, we apply the same MLP independently to each of the $H \times K$ keypoint-timestep pairs, sharing encoder weights across both temporal and spatial dimensions. Unlike approaches that require reaching a sequence of $H$ targets at predetermined timestamps, our representation only specifies that the $H$ targets be reached in the correct order. However, because we share encoder weights across both the temporal and spatial dimensions of the task specification $c$, we must explicitly inject temporal information into the representation prior to passing the encoded features into the fusion layer. To this end, we introduce a normalized positional encoding to add in the temporal information by first normalizing each timestep to the interval $[0, 1]$ and then expanding it into a high-frequency absolute positional encoding (Vaswani et al., 2017).

**Spatio-temporal attention.** After encoding $x_t$, $c$, and $u_t$, inferring future robot action trajectories from the sparse task specification $c$ requires aggregating information across neighboring keypoints in both spatial and temporal dimensions, while also capturing object-level geometry from the point cloud $x_t$. To this end, we introduce a novel Spatio-Temporal Transformer architecture outlined in Figure 2, which takes the concatenation of encoded features of $u_t$ and $c$ along the temporal axis as input tokens, while treating $x_t$ as context. Each Spatio-Temporal Attention layer in this architecture performs a three step fusion, reasoning over the sequential structure of the task (e.g., how keypoints evolve over time) and the geometric relationships within each timestep (e.g., how different keypoints and parts of the object relate spatially), which consists of: (1) a self-attention operation across the temporal dimension of the input tokens (2) a self-attention operation across the spatial dimension (3) a cross-attention operation following each self-attention operation, where the point cloud features from $x_t$ are used to ground the evolving object motion predictions into robot embodiment action presentations. This interleaved fusion scheme allows the Spatio-Temporal Transformer to incrementally refine the embeddings across layers. For example, in early layers, temporal attention (step 1) leverages $u_t$ to infer the current progress within the task specification $c$, providing a temporal anchor for downstream reasoning. In later layers, the same mechanism can focus on predicting the remaining motion trajectory by attending to how each keypoint evolves over time. Cross-attention (step 3) in early layers incorporates global geometric context from $x_t$, helping the network disambiguate or recover under-specified goals in $c$. In contrast, later layers can use cross-attention to focus on local features around the tracked keypoints $u_t$, grounding motion plans in the observed scene to support precise action generation. The output of this network is combined with robot proprioception $\rho_t$ to condition the flow-matching process for action generation.

### 3.3 Imitation Learning with Hindsight Correspondence Estimation

We propose an imitation learning paradigm to train the policy $\pi$ through self-supervision. With the goal of enabling generalization across a vast variety of scene variations and motions, we collect our training data from randomized simulated environments with diverse objects and pre-defined motion primitives. We introduce two key design options that enable our policy to be conditioned on flexible spatial correspondence specifications.

**Hindsight correspondence estimation.** To train a policy conditioned on spatial correspondence representations, we require ground-truth labels that specify how points in the scene correspond to their future locations. However, trajectories generated in our simulated environments using heuristic policies are not associated with explicit correspondence labels. Inspired by hindsight relabeling from goal-conditioned reinforcement learning (Andrychowicz et al., 2017; Fang et al., 2019b), we introduce a relabeling mechanism tailored to spatial correspondences. Specifically, at the start of each simulation episode, we identify a set of keypoints in the scene and track the ground-truth 3D locations of them throughout the episode. After the episode is complete, we relabel the achieved motion trajectories of these points in hindsight for computing the task representation during training. This automatic labeling pipeline provides a scalable and self-supervised method to obtain dense spatial correspondence labels to train COIL.

**Correspondence augmentation.** Since the correspondence labels are generated in hindsight, all keypoint locations $u_t$ in the collected training data lie exactly along the task specification trajectory $c$. However, this does not hold during deployment: keypoints can deviate from the intended task specification due to compounding factors such as policy prediction errors, external disturbances, or inaccuracies from online point tracking algorithms. To this end, we introduce a simple yet effective correspondence augmentation scheme to combat such distribution shift, and to enable the learned policy to be conditioned on task specifications of varying granularity. For each trajectory with dense 3D object flow labels of $T$ time steps and $M$ keypoints, we randomly subsample a temporal length $H \in [2, H_{\max}]$ and a keypoint count $K \in [K_{\min}, K_{\max}]$ A temporally ordered subset of $H$ time steps and $K$ keypoints is then sampled without replacement to form the spatial correspondence input for training.

We found that such a simple sub-sampling technique is enough for obtaining recovery behaviors when policy mispredicts an action or the object being manipulated is affected by external perturbation. However, to account for noise introduced by online point-tracking algorithms, we determine whether each keypoint is visible in the current camera setup during data collection, and use that information to add varying levels of Gaussian noise to the observed tracked keypoint locations $u_t$ during training.

# 4 EXPERIMENTS

We conduct experiments to evaluate the effectiveness of COIL in learning versatile robotic manipulation conditioned on spatio-temporal correspondences. Specifically, we aim to answer the following questions: (1) Can COIL learn diverse visuomotor skills generalizable to unseen objects and tasks? (2) Can the learned policy robustly ground correspondence-oriented task specifications with varying spatial and temporal granularity into effective actions? (3) How does each individual design component of COIL contribute to overall performance?

## 4.1 EXPERIMENTAL SETUP

**Environment.** While our proposed approach can be broadly applied to various platforms, we focus on table-top manipulation settings in our experiments, following the DROID benchmark (Khazatsky et al., 2024). This setting involves a 7-DoF Franka Research 3 robot equipped with a Robotiq gripper that interacts with objects on the table to achieve task success. Two external Zed 2i stereo cameras mounted on both sides of the table are used to provide RGB-D observation. We also create a simulated environment that resembles the real-world setup for training and ablation study.

**Training data.** We collect training data from randomized simulated environments containing diverse objects, containers, and tools and pre-defined motion primitives. We obtain training labels by hindsight relabeling after rolling out random motion primitives in simulation, as mentioned in Section 3.3. More details about the simulated environments and motion primitives can be found in Section A.1.

**Task design.** As shown in Figure 3, we design three real-world manipulation tasks: *Pick-and-Place*, *Sweeping*, and *Folding*. All tasks use out-of-distribution objects to rigorously test the generalization capabilities of our method and the baselines. We also create a simulation task in the Maniskill3 (Tao et al., 2024) simulator that closely resembles the *Sweeping* task to thoroughly study the key factors that enabled the success of our method. For all evaluation tasks, we vary the granularity of the task representation both spatially and temporally with three settings: Sparse, Medium, and Dense.

**Evaluation metrics.** We evaluate the performance of COIL in each manipulation task mainly using the task success rate. For the ablation study in simulation, we also report the tracking error for the successful trajectories, which quantifies the distance between the specified target positions of the keypoints and the executed object motions. Further details of the computation of these metrics can be found in the Section A.4.

## 4.2 COMPARATIVE RESULTS

**Comparative Results** We design our comparative experiments to answer whether our method can ground spatial correspondences of varying spatial and temporal granularity robustly with diverse out-of-distribution objects and manipulation tasks in a zero-shot manner. To this end, we compare

| Method | Pick-and-Place | | | Sweeping | | | Folding | | |
|---|---|---|---|---|---|---|---|---|---|
| Setting | Sparse | Medium | Dense | Sparse | Medium | Dense | Sparse | Medium | Dense |
| RT-Trajectory | 0/10 | 0/10 | 2/10 | 0/10 | 0/10 | 1/10 | 0/10 | 0/10 | 1/10 |
| General-Flow | – | – | 1/10 | – | – | 0/10 | – | – | **6/10** |
| Im2Flow2Act | 2/10 | 3/10 | 8/10 | 0/10 | 0/10 | 5/10 | 1/10 | 0/10 | 4/10 |
| **COIL (Ours)** | **8/10** | **8/10** | **9/10** | **6/10** | **6/10** | **7/10** | **6/10** | **7/10** | 6/10 |
| VLM + COIL (Ours) | – | 7/10 | – | – | 5/10 | – | – | 4/10 | – |

Table 1: **Task Performances.** We show the success rates of our method and baselines under three task specification granularity settings: *Sparse* (3–5 keypoints, 2-5 steps), *Medium* (8-12 keypoints, 16 steps), and *Dense* (32 keypoints, 32 steps). We additionally show the success rate of combining our method with a VLM to directly execute manipulation tasks given language instruction of the desired task. Across all settings, COIL consistently outperforms the baselines and shows impressive zero-shot generalization to novel manipulation tasks on the out-of-distribution *Folding* task.

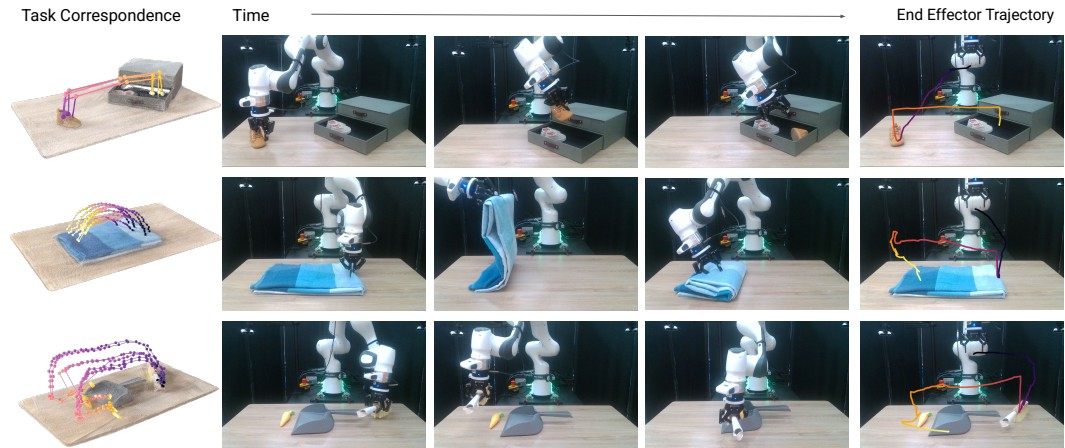

Figure 3: **Task Execution.** From left to right: selected input specification to the policy for each evaluation task, execution of our policy, and the robot's achieved end-effector trajectory. Our policy demonstrates behaviors to flexibly adapt to the objects in scene by rotating the gripper to avoid hitting the drawer in the *Pick-and-Place* task and correctly manipulating the side of the scarf in the *Folding* task. It also demonstrates accurate trajectory following capabilities in the *Sweeping* task.

our method to the following baselines, capable of performing zero-shot manipulation skills: (1) **RT-Trajectory** (Gu et al., 2023) learns a closed-loop low-level control policy conditioned on the motion of end-effector in 2D and gripper actions. We trained this baseline using our simulated data since the RT-1 dataset is not open-sourced. (2) **Im2Flow2Act** (Xu et al., 2024) trains a closed-loop control policy conditioned on 2d object flows to perform the task. Since the original work uses a different embodiment for its low-level policy, we re-trained the policy using our simulated data (3) **General Flow** (Yuan et al., 2024a) performs manipulation tasks by iteratively grounding the outputs of a short-horizon flow-generation model using heuristic iterative closest point motion planning. Similarly to the original work, we manually align the gripper prior to rolling out the model. Notably, the Sparse and Medium evaluation settings are not applicable to the General-Flow baseline, as evaluating this method requires us to iteratively query the flow-generation model in a closed-loop. We additionally report the result of **combining a VLM specification generator based on Fang et al. (2024) with our method** to show the success rates of our method when conditioned directly on language instructions.

To ensure fair comparisons across baselines, we carefully design the task interface for each baseline method. For all methods evaluated in the real world, we construct the *Sparse* and *Medium* spatial correspondence by interacting with a user interface. The *Dense* spatial correspondences are obtained by projecting 2D point tracking sequences of a human demonstration recording in 3D. We project the 3D specifications back into 2D to form inputs for the Im2Flow2Act baseline (Xu et al., 2024). For the RT-Trajectory baseline (Gu et al., 2023), we record the end-effector trajectories resulting from successful rollouts from an oracle policy and project it onto the 2D image.

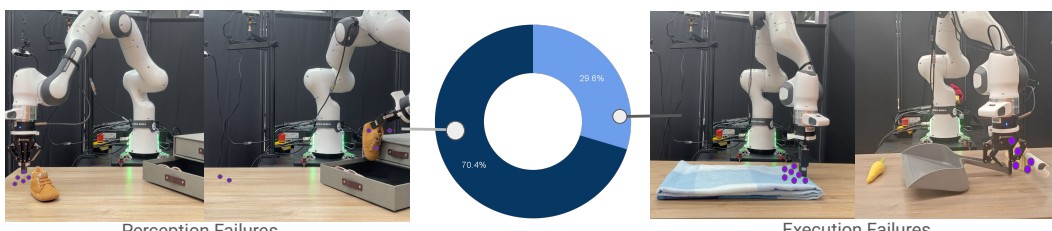

Perception Failures        Execution Failures

Figure 4: **Failure Analysis.** We categorize real-world failure cases into *perception failures* and *execution failures*, and visualize their proportions across all evaluation tasks. The majority of failures stem from inaccuracies in point tracking, particularly under occlusion or clutter. Execution failures most commonly occur during the grasping phase , often when objects are flat or lack distinctive geometry, making it difficult for the policy to localize reliable grasp points.

As shown in Table 1, COIL consistently outperforms all baselines across all tasks and spatial correspondence settings, while achieving high success rate with direct language conditioning. RT-Trajectory (Gu et al., 2023) fails on most settings, likely due to its reliance on image-conditioned policies trained entirely in simulation, which suffer from a large sim-to-real gap when deployed in the real world. Im2Flow2Act (Xu et al., 2024) shows moderate performance, but remains sensitive to the density of the input specification. The General-Flow baseline only show competitive results on the *Folding* task, as the other two tasks are out-of-distribution for their pretrained high-level flow generators. Our method demonstrates strong capability in manipulating both rigid and deformable objects, maintaining high success rates even under sparse specifications, highlighting its robustness to coarsely defined task specifications. In contrast, baseline methods exhibit significant performance degradation when the task specifications become sparse. Furthermore, the strong performance of COIL on the *Folding* task indicates its ability to generalize to novel object categories and motion dynamics, as deformable objects were not present in the training data.

### 4.3 QUALITATIVE RESULTS

**Task execution.** We visualize the task input specification and execution of our method on the real-world evaluation tasks in Figure 3. In the *Pick-and-Place* task, the policy is conditioned on a spatially and temporally sparse specification. Our policy shows emergent behaviors to flexibly adapt to surrounding objects and avoids contact with the container by rotating the gripper. In the *Folding* task, the policy is conditioned on a spatially and temporally dense specification. COIL demonstrates strong generalization to an unseen deformable object that was never encountered during training. Furthermore, despite the presence of correspondences spanning both the side and the center of the scarf, the policy correctly infers the intent to grasp from the edge, highlighting the policy's ability to interpret spatial correspondences in a space-aware manner. Across all cases, the object trajectories executed by the robot closely follow the specified target motions in 3D, validating that COIL can ground diverse spatial task specifications into coherent and purposeful visuomotor behavior.

**Motion diversity.** We observe that when provided with temporally sparse spatial correspondences, our policy produces a variety of valid trajectories that all successfully accomplish the intended task. This makes our method particularly well-suited for generating diverse successful demonstrations.

**Failure analysis.** We break down failure cases of our methods into two categories: perception failures and execution failures, and show their percentage and some examples of each failure category in Figure 4. We notice that most of the failure cases are due to errors made by the point-tracking algorithms, or noises contained in stereo depth estimations. The noise in depth can be partially mitigated by using newer stereo matching algorithms. Additionally, we observe that execution failures often occur during the grasping phase, particularly when object point clouds are heavily cluttered. In such cases, the lack of geometric distinctiveness makes it difficult for the policy to accurately infer graspable regions, resulting in failed or unstable grasps.

### 4.4 ABLATION STUDY

We conduct a thorough ablation study in a simulated environment of the *Sweeping* task under three levels of spatial and temporal granularity (Sparse, Medium, Dense) for the task specification to

| Model Variant | Success Rate (%) | Tracking Error |
|---|---|---|
| *w/o* Spatio-Temporal Attention | 60% | 0.071 |
| *w/o* Normalized PE | 74% | 0.045 |
| *w/o* Flow Randomization | 21% | 0.251 |
| *w/o* Tracked Keypoint Noise Augmentation | 54% | 0.092 |
| **COIL (Full Method)** | **82%** | **0.029** |

Table 2: **Ablation Study.** We evaluate our full method against ablation variants in the simulated *Sweeping* task and compare the average metric across Sparse (5 keypoints, 5 timesteps), Medium (8 keypoints, 12 timesteps), and Dense (32 keypoints, 32 timesteps) granularity settings of the task representation. The full COIL method consistently outperforms all variants, validating the importance of each design factor in our approach.

analyze how individual components contribute to the overall performance. Specifically, we compare our full method with four model variants: (1) *w/o Spatio-Temporal Attention*: We replace the Spatio-Temporal Transformer network in the policy architecture with 2 separate transformer encoders, each attending tokens along the temporal and spatial dimensions separately. Additionally, we separately encode the pointcloud features and the spatial correspondence features before concatenating them as the condition to diffuse actions (2) *w/o Normalized PE*: We replace our normalized position encoding layer with absolute positional encodings before feeding features into the Spatio-Temporal Transformer module. (3) *w/o Flow Randomization*: We remove the spatial and temporal sub-sampling strategy introduced in Section 3.3 (4) *w/o Tracked Keypoint Noise*: We remove the Gaussian noise augmentation strategy for tracked keypoint $u_t$ introduced in Section 3.3.

We roll out our method and its variants with 100 episodes for each task under each granularity setting, and report the average metric numbers across all three settings in Table 2. Removing spatio-temporal attention leads to a substantial increase in grasping failures, highlighting the importance of jointly reasoning over space and time to accurately interpret the task specification. Replacing normalized positional encodings results in a modest decline in success rate, but a notable increase in correspondence tracking error. Disabling flow randomization causes the most significant degradation in performance, confirming its critical role in enabling the policy to generalize across varying spatial and temporal granularities. Disabling tracked keypoint noise data augmentation causes the model to attend to tracked points that are noisy, signifying it's importance for robust action generation with noisy observations. Collectively, these results demonstrate that both architectural design and training strategies are essential for robust grounding of correspondence-based task specifications.

## 5 CONCLUSION AND DISCUSSION

We presented COIL, a conditional imitation learning framework for visuomotor control in 3D, conditioned on a novel correspondence-oriented task representation with variable spatial and temporal resolution. This formulation allows tasks to be specified directly in physical space through the intended motion of keypoints on scene objects, supporting both coarse and fine-grained specifications across diverse tasks. To robustly ground these flexible inputs into executable actions, we propose a spatio-temporal attention architecture that fuses task representations with point cloud observations and robot state. Our policy is trained using a scalable self-supervised pipeline that generates diverse demonstrations in simulation, with task specifications automatically derived via hindsight relabeling. COIL achieves strong generalization across task types, object categories, and correspondence granularities, offering a step toward scalable, adaptive, and interpretable robot learning systems.

**Limitations.** In spite of the advantages of COIL, the current method faces several limitations. First, the policy relies on accurate online keypoint tracking at test time, and existing tracking methods (Doersch et al., 2024) can introduce noise, particularly in cluttered scenes or under occlusion. Second, the method assumes task representations are externally provided and does not reason about intent ambiguity or specification quality. Extending COIL to jointly infer and refine task representations could improve autonomy and robustness. Third, while our framework operates over 3D visual inputs, it currently does not leverage other sensory modalities such as touch or force, which are critical for tasks involving contact and compliance. Incorporating multi-modal sensing into the correspondence representation is a promising direction for enabling more diverse and complex manipulation.

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

# A  APPENDIX

## A.1  DATA COLLECTION DETAILS

**Simulation Data.** To address the scarcity of real-world robot manipulation data (Mirchandani et al.; Khazatsky et al., 2024; Collaboration, 2023; Walke et al., 2023), we leverage simulation as a scalable and efficient alternative for generating diverse training trajectories. We construct a tabletop manipulation environment in the ManiSkill3 simulation platform (Tao et al., 2024), using the same Franka Research 3 robot as in our real-world setup to ensure embodiment consistency.

To promote generalization across object geometries, functionalities, and task dynamics, we curate a diverse asset set drawn from the YCB dataset (Calli et al., 2015), PartNet-Mobility dataset (Xiang et al., 2020), and publicly available 3D models. These assets are grouped into three functional categories:

- **Manipulatable Objects:** Everyday items such as fruits, bottles, legos, etc. These objects vary in geometry and texture, and serve as the primary targets for interaction.

- **Tool Objects:** Functionally extended items such as spatulas, hammers, and knives, used to enable indirect manipulation. We manually annotate each tool's functional direction and location for meaningful tool use motion generation.

- **Containers:** We include racks, boxes, baskets to enable object placement diversity for the simulated scenes.

To ensure meaningful object placement, we first sample 0–2 container objects and place them randomly on the tabletop. We then randomly sample 1–5 manipulable objects (including tools) and place each either directly on the table ($p = 0.6$) or inside one of the containers ($p = 0.4$). This setup introduces structural and spatial diversity across scenes, encouraging the policy to generalize across object arrangements, occlusions, and tool-use configurations.

Given a randomly generated environment, we introduce two types of heuristic actions to collect simulated data:

- **Random 6DoF:** We begin by selecting a random object in the scene and executing a grasp using the robot gripper. Upon successful grasping, we sample a set of 1 to 4 random waypoints $p_1, \ldots, p_w$ within the robot's workspace. A 3D cubic Bézier curve is then constructed to define a smooth end-effector trajectory passing through these waypoints. To introduce rotational variation, we randomly perturb the object's orientation at each waypoint and apply spherical linear interpolation (SLERP) between successive orientations along the trajectory. After following the full trajectory, the gripper releases the object at the final position. This procedure generates diverse motion patterns that encourage the policy to generalize across object poses and manipulation strategies.

- **Tool Use:** If a tool is present in the scene, we begin by selecting one at random and executing a grasp at its functionally annotated region, as defined by human-provided labels. Next, we select a random object on the table and move the tool such that its nearest functional point is aligned with the object. We then perform a manipulation primitive such as sweeping, poking, or hooking by translating the tool along its designated functional direction. Finally, the tool is released.

During data collection, we only include robot actions of the successful trajectories and label the spatial correspondence specification of the trajectory in hindsight, as described in the paper.

**Visual Observation Augmentation.** While we carefully select the workspace point cloud as the policy's visual observation $x_t$, a sim-to-real gap remains due to real-world visual complexities and the challenges of stereo depth estimation. To mitigate this, we augment $x_t$ during training to increase robustness to noisy depth inputs. Specifically, we add independent Gaussian noise to each point in $x_t$, and inject randomly sampled 3D points from the robot workspace to simulate spurious depth readings.

## A.2 FLOW-MATCHING HEAD

Diffusion Models (Ho et al., 2020) are powerful generative frameworks for modeling multi-modal data distributions by iteratively removing Gaussian noise, and have been widely adopted by the robotics community for modeling robot action sequences (Chi et al., 2024; Ze et al., 2024). In contrast, flow-matching approaches (Lipman et al., 2023), specifically the Rectified Flow (Liu, 2022) formulation, captures the multi-modal data distribution by modeling an optimal transport problem between a source $\mu^0$ and a target $\mu^1$ distribution, where the source distribution $\mu^0$ usually follows a standard normal distribution $N(0, 1)$.

The flow-matching head $v_\theta(\cdots|A_t^\tau, \tau, \phi)$ predicts the instantaneous velocity field that transports a noisy action trajectory $A_t^\tau$ toward the clean action trajectory under prediction timestep $\tau \in [0, 1]$ and conditioning $\phi$ (e.g., output from a feature extractor or raw observation inputs).

During training, given target action labels $A_t^1 = a_{t:t+T_a-1}$, we first sample a prediction timestep $\tau \in [0, 1]$ and interpolate between the source prior $A_t^0 \sim N(0, 1)$ and the target $A_t^1$ to obtain $A_t^\tau = (1 - \tau)A_t^0 + \tau A_t^1$. The model is then optimized to minimize the squared error between the predicted velocity and the ground-truth velocity $u_t^\tau = A_t^1 - A_t^0$:

$$\mathcal{L}_{\text{flow-matching}}(\theta) = \mathbb{E}_{A_t^0, A_t^1, \tau}\left[\|v_\theta(\cdots|A_t^\tau, \tau, \phi) - u_t^\tau\|^2\right].$$

At inference time, the learned flow field is integrated over $\tau \in [0, 1]$ using Riemann Sum with a fixed $\Delta\tau$ to transport a random sample from $A_t^0$ into the target action space, yielding diverse yet task-consistent action sequences.

**Implementation Details** The flow-matching head is modeled as a UNet architecture similar to Chi et al. (2024) but trained with the conditional flow-matching objective described above, with prediction timestep sampled from $\tau \sim \beta(1.5, 1.0)$. During inference, we set $\Delta\tau = \frac{1}{16}$.

## A.3 POINT TRACKER IMPLEMENTATION DETAILS

During inference, the tracked keypoint locations $u_t \in \mathbb{R}^{K \times 3}$ are obtained by combining a 2D online point tracking algorithm (Doersch et al., 2024) with depth estimates from stereo cameras. During environment initialization, after we obtain the task representation in the form of spatial correspondence $c$, we project each keypoint's starting 3D location $c_{0,k}$ onto the image plane **of each camera** to obtain its corresponding 2D pixel coordinates and depth value. We then compare this projected depth with the actual depth measured by the stereo cameras. A keypoint is initialized for tracking on each camera if (1) its projected pixel coordinates fall within the image bounds and (2) the measured and re-projected depths are within a predefined threshold. In each environment step, we step the 2D trackers independently for each camera to obtain (1) the 2D location of the tracked keypoint and (2) a confidence or visibility score. We select the 2D location with the highest confidence across all cameras and project it into 3D using that camera's depth estimate. This initialization and selection strategy ensures that when multiple cameras observe the same keypoint in the initial scene, and one view becomes occluded during policy rollout, tracking can still be maintained using other camera views.

## A.4 EXPERIMENT DETAILS

**Details for keypoint selection**: In our real-world experiments, keypoints were specified by uniformly sampling points on the visible surface of the target object in the observed point cloud. While task performance may vary slightly with different selections, our results show that the COIL policy is robust to this variation. In practice, we found that the density of keypoints often has a greater influence on performance than their exact placement.

**Details for each environment** Below we detail the environment set up and success metric for each evaluation task:

- **Pick-and-Place** At the beginning of each episode, a baby shoe is placed at the upper-right corner of the table. A cabinet is placed at the lower-left corner of the table and its bottom drawer is pulled open. The task is to place the shoe inside the right side of the open drawer

compartment. A trial is considered successful if (1) the shoe is correctly placed and (2) the drawer has not moved more than 2 cm during the process.

- **Sweeping** At the beginning of each episode, a brush is placed on the left side of the table, with its handle oriented at a $25°$ to $55°$ angle relative to the front edge of the table. A dustpan and a carrot are positioned to the left of the brush. The task involves lifting the brush, moving it to the side of the carrot, placing it down, and sweeping the carrot into the dustpan. A trial is considered successful if: (1) the brush is correctly picked up and repositioned beside the carrot; (2) the carrot is swept fully into the dustpan; and (3) the dustpan does not move more than 10 cm during the process.

- **Folding** At the beginning of each episode, a scarf is placed at the center of the table, with its top-bottom orientation randomized across episodes. The task is to fold the left side of the scarf over onto the right side. A trial is considered successful if: (1) the left edge of the scarf extends at least past the midpoint after folding, and (2) the folded section lies flat against the underlying surface.

For evaluations of the General Flow baseline Yuan et al. (2024a), due to the fact that it was not trained on large-scale data and cannot interpret longer, semantic instructions, we relaxed the success criterion for each of the evaluation tasks. For example, in tasks requiring semantic understanding - such as "pick and place the shoe **in the box**" - we marked a success as separately (1) picking and (2) placing the shoe, even if not in the box. For the "folding" task, General Flow exhibited common failure modes in (1) losing grasp, (2) getting stuck midway, and (3) repeatedly unfolding the towel, and everything else was marked as a success. Despite General Flow's more relaxed success criterion, COIL showed superiority on all tasks even though it was evaluated on a much stricter success metric.

## A.5 ABLATION ENVIRONMENT DETAILS

We evaluate our full method and the ablation variants in a simulated environment of the *Sweeping* task under three levels of spatial and temporal granularity (Sparse, Medium, Dense) for the task specification, with Gaussian noise added to the keypoint observation to mimic tracking noise in the real world. The task representations of the three settings are obtained by first rolling out a heuristic policy and tracking the 3D points in space, then sub-sampling both spatially and temporally.

## A.6 LLM USAGE

In preparing this manuscript, we made limited use of a large language model (ChatGPT) exclusively for polishing the writing and improving clarity of exposition. Specifically, the LLM was only used to suggest alternative phrasings for readability and conciseness.

The LLM was not involved in the conception of ideas, design of experiments, analysis of results, or any other substantive aspects of the research. All technical content, experiments, and conclusions are solely the work of the authors.

