# OpenReview forum: "Correspondence-Oriented Imitation Learning: Flexible Visuomotor Control with 3D Conditioning"
_ICLR.cc/2026/Conference — ICLR 2026 Conference Withdrawn Submission_

### Official Review · Reviewer_NLp6 · 2025-10-23

**Soundness:** 2
**Presentation:** 3
**Contribution:** 2
**Rating:** 2
**Confidence:** 4

**Summary:**

This paper introduces COIL (Correspondence-Oriented Imitation Learning), a visuomotor control framework that represents manipulation tasks as 3D trajectories of keypoints on objects, with the key innovation being **flexible spatial and temporal granularity**---unlike prior work requiring fixed numbers of keypoints or dense time sampling, COIL allows variable numbers of keypoints and timesteps to adapt task specifications from sparse pairs to dense flows. The method employs a Spatio-Temporal Transformer architecture that fuses point cloud observations with correspondence specifications through interleaved spatial and temporal attention. COIL demonstrates effectiveness on real-world manipulation tasks including rigid objects, deformable objects (some of them are never seen during training), and tool use, outperforming baselines across different specification granularities and showing generalization to novel objects and tasks.

**Strengths:**

- The paper makes important extensions to correspondence-based task specifications by
  - removing rigid assumptions about fixed keypoint counts and uniformly spaced temporal intervals, enabling specifications with variable spatial and temporal granularity.
  - requiring only that target coordinates be reached in sequential order rather than at predetermined timesteps, allowing the policy to dynamically adapt execution speed and exhibit recovery behaviors online.

- The experiments are mainly done in the real world.

**Weaknesses:**

- While Tab.1 demonstrates that COIL can execute tasks conditioned on correspondence specifications of varying granularity, most evaluated methods ***require correspondence inputs to be provided externally***. Since real-world deployment typically begins with language instructions rather than ground-truth correspondences, the paper should more thoroughly evaluate end-to-end performance with automatic correspondence generation. The current evaluation focuses primarily on execution given correspondences, leaving the practical question of how to obtain such specifications in autonomous settings underexplored.

- The VLM + COIL approach shows blank results for the Dense setting, which critically undermines the claim of supporting flexible spatial and temporal granularity. ***If the system cannot reliably generate or execute dense correspondence specifications autonomously, this represents a significant limitation for tasks requiring fine-grained motion constraints***. The paper should provide results across all granularities (Sparse, Medium, Dense) for the VLM-conditioned variant, or explicitly discuss why dense correspondence generation from language or other modalities remains an open challenge. Without demonstrating consistent performance across the full spectrum of granularities in an end-to-end setting, the flexibility claim remains incompletely validated in practice.

- While ***General Flow*** is actually a 3D-based method, the text around lines 364-365 does not clearly distinguish it from the 2D baselines (RT-Trajectory and Im2Flow2Act). Given that operating in 3D versus 2D is a fundamental architectural difference that significantly impacts depth reasoning, occlusion handling, and overall performance, the paper should explicitly clarify the dimensionality of each baseline's representation. This distinction is essential for readers to fairly interpret the comparative results and understand whether performance gains stem from the 3D representation itself or from COIL's specific architectural and training innovations.

**Questions:**

- The paper introduces flexible correspondence representations with variable numbers of keypoints (K) and timesteps (H), but does not explain how these variable-length inputs are handled during batched training. Since neural networks typically require fixed-size tensors within a batch, how do you align the spatial dimension (varying K) and temporal dimension (varying H) across different samples? Do you use padding, masking, or a dynamic batching strategy? Please clarify the implementation details for handling variable-sized correspondence tensors during training.

- Does the evaluation use a single multi-task policy trained jointly on all manipulation primitives encountered during simulation (pick-and-place, sweeping, tool use, etc.), or are separate task-specific policies trained for each evaluation scenario (one for Pick-and-Place, one for Sweeping, one for Folding)?

- Could you provide an additional ablation study removing the proprioception input ($p_t$) to the Flow-Matching Head in Fig.2? This would help clarify the contribution of robot state information versus visual and correspondence features in action generation, and whether the policy can effectively predict actions using only visual observations and task specifications.

**Details Of Ethics Concerns:**

No ethics review is needed.

---

### Official Review · Reviewer_dgED · 2025-10-25

**Soundness:** 3
**Presentation:** 2
**Contribution:** 3
**Rating:** 8
**Confidence:** 2

**Summary:**

The paper proposes an approach to learn conditional policies via imitation learning. The key contribution is a new 3D representation based on 3D points and tracks. Contextual information about the state of the world is captured by a dense 3D point cloud. The task is expressed as a sparse collection of 3D trajectories, each capturing the 3D location of a certain physical point, providing a goalpost for the robot to reach. These goalposts can be sparse and do not enforce strict timing other than temporal ordering. Finally, a 3D tracker is utilised to track the actual instantaneous location of the physical points as the robot operates, thus providing feedback on the result of the actions enacted so far. Imitation learning is used to learn a policy that takes as input the new representations as well as proprioception information and outputs actions. The policy is learned as a conditional probability distribution utilising flow matching. A mechanism to automatically generate suitable training data in simulation is also presented, adapted from prior work. This includes a small number of techniques to augment the training data and favour generalisation.

The approach is analogous to prior works that used 2D image points and flow as representation instead, and aims at demonstrating the benefits of using 3D points instead. Small-scale experiments show the potential advantages of the proposed method against prior representations.

**Strengths:**

* Extending 2D task representations to 3D is a logical step with clear potential benefits in terms of representational power.

* While the proposed representation requires specifying the desired trajectory of 3D physical points, it does so in an exceedingly prescriptive manner. Because tracks are sparse and timing other than ordering is not specified, this leaves the model and task the freedom to determine a particular sequence of actions and timing. This makes the method more applicable compared to having to specify the a more detailed plan of action.

**Weaknesses:**

* The paper could use a pass to make the math more precise and clear. For example, at line 190 the policy is a function of $o_{0:t},c$ and at line 213 as a function of $f(x_t, \rho_t, c_{t:H})$. What is $\rho$? Is that supposed to be $u$? Why is $c$ indexed in that manner? Is there any other proprioception information such as joint angles?

* If I understand the mechanism for generating training data in Section 3.3, the idea is to simulate first episodes by randomising the robot actions, and then reinterpret these sequences as examples of control. How likely is it that this would generate meaningful trajectories that are useful for the robot to learn from?

Minor:

* Line 461: "it's" -> "its"

* I did not find supplementary materials/videos with the paper. The latter in particular could have been informative, and could have been used to illustrate the effectiveness of or lack thereof of the approach.

**Questions:**

The topic of this paper is a bit beyond my typical purview, and while I have done my best to offer a balanced view above, I would defer to more “in-topic" reviewers to asses certain aspects of the paper, especially concerning novelty and significance to the field.

Please address the questions raised in the "weaknesses" box above.

---

### Official Review · Reviewer_QFh6 · 2025-10-30

**Soundness:** 2
**Presentation:** 2
**Contribution:** 2
**Rating:** 4
**Confidence:** 3

**Summary:**

The paper introduces a novel correspondence-oriented task representation with variable spatial and temporal resolutions for policy learning in 3D. The formulation enables task specification in the 3D physical spaces, supporting both coarse and fine-grained specifications. The paper evaluates the proposed method on 3 tasks under sparse, medium, and dense task specification settings. The paper also includes ablation studies to justify the design choices made in the paper.

**Strengths:**

- The paper attempts to address an important problem of task specification for conditioning robot policies.
- The authors compare the proposed method on 3 real world tasks across sparse, medium, and dense task specification settings. COIL outperforms baselines in all these settings.
- The paper includes an ablation study to justify the design choices in COIL.
- The paper also includes qualitative results which give the reader a better understanding of the working of the method and its failure modes.
- The paper includes a limitations section.

**Weaknesses:**

Including both weaknesses as well as questions tied to the weaknesses below.
- How are the correspondence representations specified during training and at inference? How are the keypoints on each object determined and how is the trajectory of keypoints obtained? Once keypoints on each object are detected in the first frame, I believe they can be tracked across the trajectory to get the keypoints for the whole trajectory in the training data. However, how are they obtained during inference when there is no access to future timesteps? More details about generating the correspondence representations will be much appreciated.
- How are target steps c obtained? Is it the H future points from a given timestep with points selected at certain intervals? Or is it trajectory subsampled to H steps?
- The proposed method seems to be a 3D variant of RT-Trajectory. From what I understand, RT-Trajectory and Im2Flow2Act are 2D baselines and General Flow while 3D has a pretrained flow generation model which is out-of-domain for certain task settings and objects. Based on this, the baselines in the paper might not be appropriate for the evaluations. There have been several 3D keypoint based policy learning papers of late which could be considered instead [1,2,3]. Comparing with a 3D baseline will help disambiguate whether the performance boost is coming from the 3D representations or something about the task specification.

[1] Huang, Wenlong, et al. "Rekep: Spatio-temporal reasoning of relational keypoint constraints for robotic manipulation." arXiv preprint arXiv:2409.01652 (2024).
[2] Zhu, Yifeng, et al. "Vision-based manipulation from single human video with open-world object graphs." arXiv preprint arXiv:2405.20321 (2024).
[3] Haldar, Siddhant, and Lerrel Pinto. "Point policy: Unifying observations and actions with key points for robot manipulation." arXiv preprint arXiv:2502.20391 (2025).

**Questions:**

It would be great if the authors could address questions in the weaknesses section.

---

### Official Review · Reviewer_pUc5 · 2025-11-01

**Soundness:** 3
**Presentation:** 3
**Contribution:** 3
**Rating:** 6
**Confidence:** 4

**Summary:**

The paper introduces COIL (Correspondence-Oriented Imitation Learning), a visuomotor control framework conditioned on 3D spatio-temporal correspondences of keypoints. Instead of fixed numbers of keypoints or uniformly sampled trajectories, COIL allows variable spatial (K keypoints) and temporal (H steps) granularity, then uses a spatio-temporal Transformer that fuses task specifications, tracked keypoints, point clouds, and proprioception, with a flow-matching head for multi-step action prediction. Training uses self-supervision in simulation via hindsight correspondence relabeling and sub-sampling/noise augmentation to handle sparse specs and tracking noise. On three real-world tasks (Pick-and-Place, Sweeping, Folding) and a simulator variant, COIL reportedly outperforms several flow/trajectory-conditioned baselines, with ablations highlighting the importance of spatio-temporal attention, positional encodings, sub-sampling (“flow randomization”), and noise augmentation.

**Strengths:**

1.Flexible task interface: Clean formulation of 3D correspondence specs that support variable K and H, bridging between sparse start–goal and dense flows within one interface. This is clearly articulated and practically useful.

2.Architectural design: The interleaved temporal-self, spatial-self, and point-cloud cross-attention is a thoughtful way to ground sparse plans into precise actions; the role of normalized temporal P.E. is well motivated.

3.Scalable training recipe: Hindsight correspondence estimation + sub-sampling (“flow randomization”) + tracked-keypoint noise form a coherent, self-supervised pipeline that targets deployment realities (sparse specs, tracking errors).

4.Empirical gains & clarity: On three tabletop tasks with OOD objects, COIL shows consistent gains over baselines; ablations quantify each component’s contribution (notably large drop without flow randomization).

**Weaknesses:**

1.Baseline coverage/fairness: While 2D-flow and end-effector baselines are included, comparisons omit strong 3D point-cloud policies (e.g., recent 3D diffusion/point-conditioned policies also cited by the paper), which would better isolate the benefit of the correspondence interface vs. modern 3D observation encoders. Additionally, General-Flow is only applicable in the dense setting and RT-Trajectory is retrained on the authors’ data, leaving open questions of tuning parity.

2.Reliance on external tracking at test time: Performance depends on reliable online keypoint tracking; many failures stem from tracking under occlusion/clutter. It’s unclear how sensitive COIL is to tracker drift, identity swaps, or depth noise distributions outside ZED setups.

**Questions:**

1.Tracker robustness: Which point-tracking model is used online, and how does performance degrade under occlusion, identity switches, and depth noise? Could you provide a stress test varying tracker quality and quantify its effect on success rate?

2.Baselines with 3D policies: Can you compare against 3D point-cloud-native policies (e.g., 3D diffusion or point-conditioned policies) trained on the same simulator data to disentangle the benefit of correspondences vs. a strong 3D observation backbone?

3.Specification authoring: For sparse specs, how much time does a user need to place K keypoints and H steps? Any human factors study on inter-annotator variance (whether different operators specify comparable correspondences)?

---

### Note · Authors · 2025-12-04

I have read and agree with the venue's withdrawal policy on behalf of myself and my co-authors.